# Layer Sparsity in Neural Networks

## Abstract

Sparsity has become popular in machine learning, because it can save computational resources, facilitate interpretations, and prevent overfitting. In this paper, we discuss sparsity in the framework of neural networks. In particular, we formulate a new notion of sparsity that concerns the networks' layers and, therefore, aligns particularly well with the current trend toward deep networks. We call this notion layer sparsity. We then introduce corresponding regularization and refitting schemes that can complement standard deep-learning pipelines to generate more compact and accurate networks.

## 1 Introduction

The number of layers and the number of nodes in each layer are arguably among the most fundamental parameters of neural networks. But specifying these parameters can be challenging: deep and wide networks, that is, networks with many layers and nodes, can describe data in astounding detail, but they are also prone to overfitting and require large memory, CPU, energy, and so forth. The resource requirements can be particularly problematic for real-time applications or applications on fitness trackers and other wearables, whose popularity has surged in recent years. A promising approach to meet these challenges is to fit networks sizes adaptively, that is, to allow for many layers and nodes in principle, but to ensure that the final network is "simple" in that it has a small number of connections, nodes, or layers (Changpinyo et al., 2017; Han et al., 2016; Kim et al., 2016; Liu et al., 2015; Wen et al., 2016).

Popular ways to fit such simple and compact networks include successively augmenting small networks (Ash, 1989; Bello, 1992), pruning large networks (Simonyan & Zisserman, 2015), or explicit sparsity-inducing regularization of the weight matrices, which we focus on here. An example is the $\ell_1$-norm, which can reduce the number of connections. Another example is the $\ell_1$-norm grouped over the rows of the weight matrices, which can reduce the number of nodes. It has been shown that such regularizers can indeed produce networks that are both accurate and yet have a small number of nodes and connections either in the first layer (Feng & Simon, 2017) or overall (Alvarez & Salzmann, 2016; Liu et al., 2015; Scardapane et al., 2017). Such sparsity-inducing regularizers also have a long-standing tradition and thorough theoretical underpinning in statistics (Hastie et al., 2015).

But while sparsity on the level of connections and nodes has been studied in some detail, sparsity on the level of layers is much less understood. This lack of understanding contrasts the current trend to deep network architectures, which is supported by state-of-the-art performances of deep networks (LeCun et al., 2015; Schmidhuber, 2015), recent approximation theory for ReLU activation networks (Liang & Srikant, 2016; Telgarsky, 2016; Yarotsky, 2017), and recent statistical theory (Golowich et al., 2017; Kohler & Langer, 2019; Taheri et al., 2020). Hence, a better understanding of sparsity on the level of layers seems to be in order.

Therefore, we discuss in this paper sparsity with a special emphasis on the networks' layers. Our key observation is that for typical activation functions such as ReLU, a layer can be removed if all its parameter values are non-negative. We leverage this observation in the development of a new regularizer that specifically targets sparsity on the level of layers, and we show that this regularizer can lead to more compact and more accurate networks.

Our three main contributions are:

1. We introduce a new notion of sparsity that we call *layer sparsity*.
2. We introduce a corresponding regularizer that can reduce network sizes.
3. We introduce an additional refitting step that can further improve prediction accuracies.

In Section 2, we specify our framework, discuss different notions of sparsity, and introduce our refitting scheme. In Section 3, we establish a numerical proof of concept. In Section 4, we conclude with a discussion.

## 2 SPARSITY IN NEURAL NETWORKS

We first state our framework, then discuss different notions of sparsity, and finally introduce a refitting scheme.

### 2.1 MATHEMATICAL FRAMEWORK

To fix ideas, we first consider fully-connected neural networks that model data according to

$$y_i = \boldsymbol{f}^1 \Big[ W^1 \boldsymbol{f}^2 \big[ ... \boldsymbol{f}^l [W^l \boldsymbol{x}_i] \big] \Big] + u_i \ , \tag{1}$$

where $i \in \{1, \ldots, n\}$ indexes the $n$ different samples, $y_i \in \mathbb{R}$ is the output, $\boldsymbol{x}_i \in \mathbb{R}^d$ is the corresponding input with $d$ the input dimension, $l$ is the number of layers, $W^j \in \mathbb{R}^{p_j \times p_{j+1}}$ for $j \in \{1, \ldots, l\}$ are the weight matrices with $p_1 = 1$ and $p_{l+1} = d$, $\boldsymbol{f}^j : \mathbb{R}^{p_j} \to \mathbb{R}^{p_j}$ for $j \in \{1, \ldots, l\}$ are the activation functions, and $u_i \in \mathbb{R}$ is the random noise. Extensions beyond fully-connected networks are straightforward—see Section 2.5.

We summarize the parameters in $\boldsymbol{W} := (W^1, \ldots, W^l) \in \mathcal{V} := \{\boldsymbol{V} = (V^1, \ldots, V^l) : V^j \in \mathbb{R}^{p_j \times p_{j+1}}\}$, and we write for ease of notation

$$f_{\boldsymbol{V}}[\boldsymbol{x}_i] := \boldsymbol{f}^1 \Big[ V^1 \boldsymbol{f}^2 \big[ ... \boldsymbol{f}^l [V^l \boldsymbol{x}_i] \big] \Big] \tag{2}$$

for $\boldsymbol{V} \in \mathcal{V}$.

Neural networks are usually fitted based on regularized estimators in Lagrange

$$\widehat{\boldsymbol{W}} \in \underset{\boldsymbol{V} \in \mathcal{V}}{\operatorname{argmin}} \big\{ \text{DataFit}[y_1, \ldots, y_n, \boldsymbol{x}_1, \ldots, \boldsymbol{x}_n] + h[\boldsymbol{V}] \big\} \tag{3}$$

or constraint form

$$\widehat{\boldsymbol{W}} \in \underset{\substack{\boldsymbol{V} \in \mathcal{V} \\ h[\boldsymbol{V}] \leq 1}}{\operatorname{argmin}} \big\{ \text{DataFit}[y_1, \ldots, y_n, \boldsymbol{x}_1, \ldots, \boldsymbol{x}_n] \big\} \ , \tag{4}$$

where $\text{DataFit} : \mathbb{R}^n \times \mathbb{R}^{n \times d}$ is a data-fitting function such as least-squares $\sum_{i=1}^n (y_i - f_{\boldsymbol{V}}[\boldsymbol{x}_i])^2$, and $h : \mathcal{V} \to [0, \infty)$ is a regularizer such as the elementwise $\ell_1$-norm $\sum_{j,k,l} |(V^j)_{kl}|$. We are particularly interested in regularizers that induce sparsity.

### 2.2 STANDARD NOTIONS OF SPARSITY

We first state two regularizers that are known in deep learning and the corresponding notions of sparsity.

**Connection sparsity**   Consider the vanilla $\ell_1$-regularizer

$$h^{\text{C}}[\boldsymbol{V}] := \sum_{j=1}^l (r^{\text{C}})_j \|V^j\|_1 := \sum_{j=1}^l (r^{\text{C}})_j \sum_{v=1}^{p_j} \sum_{w=1}^{p_{j+1}} |(V^j)_{vw}| \ ,$$

where $\boldsymbol{r}^{\text{C}} \in [0, \infty)^l$ is a vector of tuning parameters. This regularizer is the deep learning equivalent of the lasso regularizer in linear regression (Tibshirani, 1996) and has received considerable attention

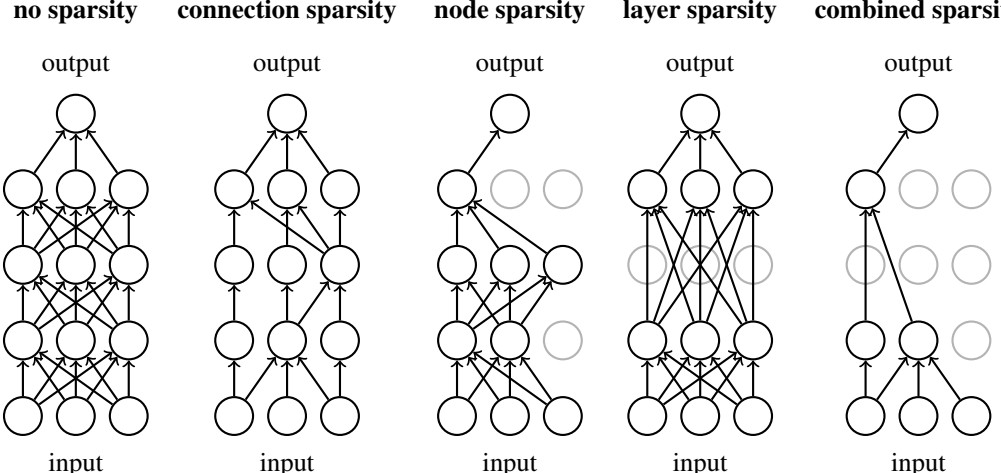

Figure 1: feedforward neural networks with different types of sparsity

recently (Barron & Klusowski, 2018; 2019; Kim et al., 2016). The regularizer acts on each individual connection, pruning a full network (first network from the left in Figure 1) to a more sparsely connected network (second network in Figure 1). We, therefore, propose to speak of *connection sparsity.*

**Node sparsity**    Consider a grouped version of the above regularizer

$$h^{\mathrm{N}}[\boldsymbol{V}] := \sum_{j=1}^{l}(r^{\mathrm{N}})_j \|V^j\|_{2,1} := \sum_{j=1}^{l}(r^{\mathrm{N}})_j \sum_{v=1}^{p_j}\sqrt{\sum_{w=1}^{p_{j+1}}|(V^j)_{vw}|^2} \ ,$$

where $\boldsymbol{r}^{\mathrm{N}} \in [0,\infty)^l$ is again a vector of tuning parameters. This regularizer is the deep learning equivalent of the group lasso regularizer in linear regression (Bakin, 1999) and has received some attention recently (Alvarez & Salzmann, 2016; Feng & Simon, 2017; Scardapane et al., 2017). The regularizer acts on all connections that go into a node simultaneously, rendering entire nodes inactive (third network in Figure 1). We, therefore, propose to speak of *node sparsity.*

### 2.3   LAYER SPARSITY

We now complement the two existing regularizers and notions of sparsity with a new, third notion.

**Layer sparsity**    Consider the regularizer

$$h^{\mathrm{L}}[\boldsymbol{V}] := \sum_{j=1}^{l-1}(r^{\mathrm{L}})_j \|V^j\|_{2,+} := \sum_{j=1}^{l-1}(r^{\mathrm{L}})_j\sqrt{\sum_{v=1}^{p_j}\sum_{w=1}^{p_{j+1}}\big(\mathrm{neg}[(V^j)_{vw}]\big)^2} \ , \tag{5}$$

where $\boldsymbol{r}^{\mathrm{L}} \in [0,\infty)^{l-1}$ is a vector of tuning parameters, and $\mathrm{neg}[a] := \min\{a,0\}$ is the negative part of a real value $a \in \mathbb{R}$. This regularizers does not have an equivalent in linear regression, and it is also new in deep learning.

We argue that the regularizer can give rise to a new type of sparsity. The regularizer can be disentangled along the layers according to

$$h^{\mathrm{L}}[\boldsymbol{V}] = \sum_{j=1}^{l-1}(r^{\mathrm{L}})_j h^{\mathrm{L},j}[V^j]$$

with

$$h^{\mathrm{L},j}[V^j] := \sqrt{\sum_{v=1}^{p_j}\sum_{w=1}^{p_{j+1}}\big(\mathrm{neg}[(V^j)_{vw}]\big)^2} \qquad \text{for } j \in \{1,\dots,l-1\} \ .$$

We then focus on an individual inner layer, that is, a layer that corresponds to an index $j \in \{2, \ldots, l - 1\}$. To fix ideas, we consider the popular ReLU activation (Glorot et al., 2011; Hahnloser, 1998; Hahnloser et al., 2000; Salinas & Abbott, 1996); in other words, we consider $(\boldsymbol{f}^j)_q[t] := f_{\mathrm{ReLU}}[t] := \max\{t, 0\}$ for all $j \in \{2, \ldots, l\}$, $q \in \{1, \ldots, p_j\}$, and $t \in \mathbb{R}$ (the activation of the output layer can be arbitrary). It is now easy to show that the regularizer $h^{\mathrm{L}}$ indeed induces sparsity on the level of layers.

**Theorem 1** (Layer Sparsity). *Consider $j \in \{2, \ldots, l - 1\}$, and define a merged weight matrix as $V^{j-1,j} := V^{j-1}V^j \in \mathbb{R}^{p_{j-1} \times p_{j+1}}$. It holds that*

$$h^{\mathrm{L},j}[V^j] = 0 \quad \Rightarrow \quad \boldsymbol{f}^{j-1}\big[V^{j-1}\boldsymbol{f}^j[V^j\boldsymbol{z}]\big] = \boldsymbol{f}^{j-1}\big[V^{j-1,j}\boldsymbol{z}\big] \quad \text{for all } \boldsymbol{z} \in [0, \infty)^{p_{j+1}} .$$

*Proof of Theorem 1.* If $h^{\mathrm{L},j}[V^j] = 0$, then $(V^j)_{qm} \geq 0$ for all $q \in \{1, \ldots, p_j\}$, $m \in \{1, \ldots, p_{j+1}\}$. Hence, it holds for all $q \in \{1, \ldots, p_j\}$ that $(V^j\boldsymbol{z})_q \geq 0$ and, therefore, that $\boldsymbol{f}^j[V^j\boldsymbol{z}] = V^j\boldsymbol{z}$. The theorem follows then by the fact that $V^{j-1,j} = V^{j-1}V^j$. □

A key property of the ReLU function is positive homogeneity. That positive homogeneity can allow for moving weights between layers had been observed in Barron & Klusowski (2018); here, we use the positive homogeneity to merge layers. The idea is as follows: $h^{\mathrm{L},j}[V^j] = 0$ means in view of the stated theorem that we can redefine the network of depth $l$ as a network of depth $l - 1$ by removing the function $\boldsymbol{f}^j$, replacing the weights $V^{j-1}$ by $V^{j-1,j}$, and then removing the $j$th layer altogether.

Theorem 1 can be applied sequentially to neighboring layers; hence, the regularization can merge not only one but many layers into one. In conclusion, our new regularizer $h^{\mathrm{L}}$ acts on all nodes and connections of each layer simultaneously, rendering entire layers inactive (fourth network in Figure 1). We, therefore, propose to speak of *layer sparsity*.

The concept of layer sparsity and Theorem 1 in particular do not hinge on the exact choice of the regularizer in (5): one can take any function $h^{\mathrm{L}}$ that can be disentangled along the layers as described and that ensures the fact that $h^{\mathrm{L},j}[V^j] = 0$ implies $\min_{k,l}(V^j)_{kl} \geq 0$.

We illustrate layer sparsity with two examples.

**Example 1** (Identity Activation). *We first highlight the meaning of layer sparsity in a simplistic setting that does not rely on Theorem 1. We consider identity activation, that is, $(\boldsymbol{f}^j)_q[t] = t$ for all $j \in \{2, \ldots, l\}$, $q \in \{1, \ldots, p_j\}$, and $t \in \mathbb{R}$. The networks in (2) can then be written as*

$$f_{\boldsymbol{V}}[\boldsymbol{x}_i] = \boldsymbol{f}^1[V^1 \cdots V^l \boldsymbol{x}_i] .$$

*In other words, the initial $l$-layer network can be compressed into a one-layer network with activation function $\boldsymbol{f}^1$ and parameter matrix $V^1 \cdots V^l \in \mathbb{R}^{1 \times d}$. This setting with identity activation is, of course, purely academic, but it motivates an important question: can parts of networks be compressed similarly in the case of ReLU?*

*Theorem 1 gives an answer to this question: if $h^{\mathrm{L},j}[V^j] = 0$, then the $j$th and $(j - 1)$th layers can be combined. In the extreme case $h^{\mathrm{L},2}[V^2] = \cdots = h^{\mathrm{L},l}[V^l] = 0$ and non-negative input, the network can be condensed into a one-layer network just as in the linear case. In this sense, one can understand our layer regularizer is as a measure for the networks' "distance to linearity." We detail this further in the following example.*

**Example 2** (ReLU Activation). *We now illustrate how layer sparsity compresses and, therefore, simplifies networks in the case of ReLU activation. We fix an initial network $f_{\boldsymbol{N}}$ parameterized by $\boldsymbol{N} \in \mathcal{V}$. We identify the active layers of the network by*

$$\mathcal{S} \equiv \mathcal{S}[\boldsymbol{N}] := \big\{ j \in \{2, \ldots, l - 1\} \, : \, h^{\mathrm{L},j}[N^j] \neq 0 \big\} \cup \{1, l\} . \tag{6}$$

*Thus, $\mathcal{S}$ and $\{1, \ldots, l\} \setminus \mathcal{S}$ contain the indexes of the relevant and irrelevant layers, respectively. (We always consider the input and output layers as active.) The level of sparsity, that is, the number of active layers, is $s := |\mathcal{S}| \leq l$.*

*Observe first that the theorem's restriction to $\boldsymbol{z}$'s that have non-negative elements makes sense: by the definition of $f_{\mathrm{ReLU}}$, the outputs of every ReLU layer are non-negative. We now denote the indexes in $\mathcal{S}$ in an orderly fashion: $j_1, \ldots, j_s \in \mathcal{S}$ such that $j_1 < \cdots < j_s = l$. We then define scaled*

*versions of the corresponding merged matrices: if $j_i - 1 \in \mathcal{S}$ or $j_i \in \{1, l\}$, we do the "trivial merge" $M^{j_i} := N^{j_i} \in \mathbb{R}^{p_{j_i} \times p_{j_i+1}}$; otherwise, we do the "non-trivial merge"*

$$M^{j_i} := N^{j_{i-1}+1} \cdots N^{j_i} \in \mathbb{R}^{p_{j_{i-1}+1} \times p_{j_i+1}} \ .$$

*In other words, we merge all irrelevant layers between the $j_{i-1}$th and $j_i$th layers into the $j_i$th layer.*

*We can then compress the data-generating model in* (1) *into*

$$y_i = \boldsymbol{f}^{j_1}\left[M^{j_1}\boldsymbol{f}^{j_2}\left[...\boldsymbol{f}^{j_s}[M^{j_s}\boldsymbol{x}_i]\right]\right] + u_i$$

*with* $\boldsymbol{M} := (M^{j_1}, \ldots, M^{j_s}) \in \mathcal{V}_{\mathcal{S}} := \{\boldsymbol{V} = (V^1, \ldots, V^s) : V^i \in \mathbb{R}^{p_{j_{i-1}+1} \times p_{j_i+1}}\}$. *Formulated differently, we can condense the original network according to*

$$f_{\boldsymbol{N}}[\boldsymbol{x}_i] = f_{\boldsymbol{M}}[\boldsymbol{x}_i] = \boldsymbol{f}^{j_1}\left[M^{j_1}\boldsymbol{f}^{j_2}\left[...\boldsymbol{f}^{j_s}[M^{j_s}\boldsymbol{x}_i]\right]\right] \ ,$$

*that is, we can formulate the initial ReLU activation network with $l$ layers as a new ReLU activation network with $s$ layers.*

*The new network is still a ReLU activation network but has a smaller number of layers if $s < l$ and, consequently, a smaller number of parameters in total: the total number of parameters in the initial network is $\sum_{j=1}^{l}(p_j \times p_{j+1})$, while the total number of parameters in the transformed network is only $\sum_{i=1}^{s}(p_{j_{i-1}+1} \times p_{j_i+1})$.*

Our concept for regularizing layers is substantially different from existing ones: our layer-wise regularizer induces weights to be *non-negative*, whereas existing layer-wise regularizers induce weights to be *zero* (Wen et al., 2016, Section 3.3). The two main advantages of our approach are that it (i) does not require shortcuts to avoid trivial networks and (ii) does not implicitly enforce connection or node sparsity. We thus argue that our layer sparsity is a much more natural and appropriate way to capture and regularize network depths.

Layer sparsity more closely relates to ResNets (He et al., 2016). The recent popularity of ResNets is motivated by two observations: 1. Solvers seem to struggle with finding good minima of deep networks; even training accuracies can deteriorate when increasing the number of layers. 2. Allowing for linear mappings that short-circuit parts of the network seem to help solvers in finding better minima. From our viewpoint here, one can argue that ResNets use these linear mappings to regulate network depths adaptively and, therefore, are related to layer sparsity. But importantly, ResNets are even more complex than the networks they are based on, while our notion simplifies networks.

Since, as one can verify again readily, all three regularizers are convex, any combination of them is also convex. Such combinations can be used to obtain networks that are sparse in two or all three aspects (last network in Figure 1). In this sense, the different notions of sparsity are not competing but rather complementing each other.

## 2.4 REFITTING

Sparse networks can be used directly, but they can also be a basis for further optimization: one can adjust the network architecture according to the non-zero pattern of the sparse network and then re-estimate the parameters of this smaller network. Such strategies are well-known in statistics and machine learning under the name *refitting* (Lederer, 2013; Chzhen et al., 2019). The theoretical underpinning of these strategies is the insight that regularization creates a bias that—in certain cases—can be alleviated by a subsequent "unbiasing" step. In deep learning, refitting has been studied recently under the name *lottery-ticket hypothesis* (Frankle & Carbin, 2019).

We now formulate a version of refitting for layer sparsity. We stay in the framework of Example 2 to keep the notation light.

**Example 3** (ReLU Activation Cont.)**.** *Consider the model in* (1) *with the specifications of Example 2, and consider a corresponding layer-sparse estimator $\widehat{\boldsymbol{W}}$ of the parameters such as* (3) *with $h = h^{\mathrm{L}}$. In line with* (6)*, we denote the set of the active layers by*

$$\mathcal{S} = \mathcal{S}[\widehat{\boldsymbol{W}}] = \left\{ j \in \{1, \ldots, l-1\} \ : \ h^{\mathrm{L},j}[\widehat{W}^j] \neq 0 \right\} \cup \{1, l\}$$

*and the corresponding parameter space of the condensed network by $\mathcal{V}_\mathcal{S} \equiv \mathcal{V}_\mathcal{S}[\widehat{\boldsymbol{W}}] = \{\boldsymbol{V} = (V^1, \ldots, V^s) : V^i \in \mathbb{R}^{p_{j_{i-1}}+1 \times p_{j_i}+1}\}$, where $s := |\mathcal{S}|$. The least-squares refitted estimator for the parameters in the condensed network is then*

$$\widehat{\boldsymbol{W}}_\mathcal{S} \in \underset{\boldsymbol{V} \in \mathcal{V}_\mathcal{S}}{\operatorname{argmin}} \left\{ \sum_{i=1}^n \left(y_i - f_{\boldsymbol{V}}[\boldsymbol{x}_i]\right)^2 \right\} \ . \tag{7}$$

*Hence, the estimator $\widehat{\boldsymbol{W}}$ complemented with least-squares refitting yields the network*

$$f_{\widehat{\boldsymbol{W}}_\mathcal{S}}[\boldsymbol{x}_i] = \boldsymbol{f}^{j_1}\left[\widehat{W}_\mathcal{S}^{j_1} \boldsymbol{f}^{j_2}\left[...\boldsymbol{f}^{j_s}\left[\widehat{W}_\mathcal{S}^{j_s} \boldsymbol{x}_i\right]\right]\right] \ . \tag{8}$$

This strategy corresponds to the "one-shot approach" in Frankle & Carbin (2019); one could extend it along the lines of their iterative approach, but the numerical results indicate that this is not necessary. Also, in contrast to the results in Frankle & Carbin (2019), our results indicate that keeping the initialization is not necessary either.

## 2.5 Extensions Beyond Fully-Connected Networks

We have illustrated our ideas with feedforward networks that have fully connected layers, but the principles of layer sparsity apply much more generally. Consider a fixed hidden layer with index $j \in \{2, \ldots, l-1\}$. In the fully-connected networks (2), this layer corresponds to a function $\boldsymbol{z} \mapsto \boldsymbol{f}^j[V^j \boldsymbol{z}]$ with weights $V^j \in \mathbb{R}^{p_j \times p_{j+1}}$. We now generalize these functions to

$$\boldsymbol{z} \mapsto \boldsymbol{f}^j\left[\sum_{k=1}^m V^{j,k} \boldsymbol{z} + \boldsymbol{b}^j\right]$$

with weights $\boldsymbol{V}^j := (V^{j,1}, \ldots, V^{j,m}) \in \mathcal{M}^{j,1} \times \cdots \times \mathcal{M}^{j,m}$ and bias $\boldsymbol{b}^j \in \mathcal{B}^j$, and with arbitrary nonempty subsets $\mathcal{M}^{j,1} \times \cdots \times \mathcal{M}^{j,m} \subset \mathbb{R}^{p_j \times p_{j+1}}$ and $\mathcal{B}^j \subset \mathbb{R}^{p_j}$. The corresponding layer-sparse regularizer is then

$$h^{\mathrm{L},j}[\boldsymbol{V}^j] := \sqrt{\sum_{v=1}^{p_j}\sum_{w=1}^{p_{j+1}}\sum_{k=1}^m \left(\mathrm{neg}[(V^{j,k})_{vw}]\right)^2 + \sum_{u=1}^{p_j}\left(\mathrm{neg}[(\boldsymbol{b}^j)_u]\right)^2} \ .$$

We can confirm immediately that this regularizer has the same effect as its analog in the fully-connected case. Indeed, under the assumptions of Theorem 1, it holds that

$$h^{\mathrm{L},j}[\boldsymbol{V}^j] = 0$$
$$\Rightarrow \quad \boldsymbol{f}^{j-1}\left[V^{j-1}\boldsymbol{f}^j\left[\sum_{k=1}^m V^{j,k}\boldsymbol{z} + \boldsymbol{b}^j\right]\right] = \boldsymbol{f}^{j-1}[V^{j-1,j}\boldsymbol{z} + \boldsymbol{b}^{j-1,j}] \quad \text{for all } \boldsymbol{z} \in [0,\infty)^{p_{j+1}} \ ,$$

where $V^{j-1,j} := \sum_{k=1}^m V^{j-1}V^{j,k} \in \mathbb{R}^{p_{j-1} \times p_{j+1}}$ and $\boldsymbol{b}^{j-1,j} := V^{j-1}\boldsymbol{b}^j \in \mathbb{R}^{p_{j-1}}$. In other words, if the value of the regularizer is zero, the $j$th layer can be merged into the $(j-1)$th layer as before.

These observations highlight the fact that layer sparsity applies very generally: it only requires the properties of ReLU-type activations and the linearities that exist within most types of layers. As a concrete application, layer sparsity can compress networks that have convolutional layers, where $m$ specifies the number of feature maps and $\mathcal{M}^{j,m}$ the non-zero patterns of the filters.

## 3 Empirical Study

We now confirm in a brief empirical study the fact that layer regularization can 1. improve prediction accuracies and 2. reduce the number of active layers.

### 3.1 Artificial Data

We start with artificial data.

| | $d = 2, n = 100$ | | | | | | | | | |
|---|---|---|---|---|---|---|---|---|---|---|
| | $l - 1 = 10$ | | | | | | $l - 1 = 25$ | | | |
| | $s_W = 0.1$ | | $s_W = 0.3$ | | $s_W = 0.9$ | | $s_W = 0.1$ | | $s_W = 0.3$ | |
| Method | $\widehat{\text{mse}}$ | $\hat{s}$ | $\widehat{\text{mse}}$ | $\hat{s}$ | $\widehat{\text{mse}}$ | $\hat{s}$ | $\widehat{\text{mse}}$ | $\hat{s}$ | $\widehat{\text{mse}}$ | $\hat{s}$ |
| LS | 1.123 (1.396) | 10 (0) | 1.112 (1.448) | 10 (0) | 1.060 (1.349) | 10 (0) | 0.989 (1.263) | 25 (0) | 0.988 (1.260) | 25 (0) |
| **SLS** | 0.018 (0.023) | 10 (0) | 0.035 (0.616) | 10 (0) | 0.203 (0.830) | 10 (0) | 0.142 (1.020) | 25 (0) | 0.147 (0.985) | 25 (0) |
| ILS | 0.006 (0.011) | 1 (2) | 0.008 (0.023) | 3 (4) | 0.907 (1.272) | 9 (10) | 0.003 (0.024) | 2 (4) | 0.050 (1.012) | 7 (9) |
| **FLS** | 0.007 (0.013) | 1 (1) | 0.008 (0.128) | 2 (4) | 0.364 (1.137) | 8 (9) | 0.009 (0.985) | 2 (5) | 0.021 (1.120) | 4 (13) |

Table 1: encouraging layer sparsity can reduce the prediction error and the model complexity

### 3.1.1 SIMULATION FRAMEWORK

We generate data according to the model in (2). The most outside activation function $\boldsymbol{f}^1$ is the identity function, and the coordinates of all other activation functions $\boldsymbol{f}^2, \ldots, \boldsymbol{f}^l$ are ReLU functions. The input vectors $\boldsymbol{x}_1, \ldots, \boldsymbol{x}_n$ are jointly independent and standard normally distributed in $d$ dimensions; the noise random variables $u_1, \ldots, u_n$ are independent of the input, jointly independent, and standard normally distributed in one dimension. For a given sparsity level $s_W \in [0, 1]$, a vector $\boldsymbol{s} \in \{0, 1\}^{l-1}$ with independent Bernoulli distributed entries that have success parameter $s_W$ is generated. The entries of the parameter matrix $W^1$ are sampled independently from the uniform distribution on $(-2, 2)$, and the entries of the parameter matrices $W^2, \ldots, W^l$ are sampled independently from the uniform distribution on $(0, 2)$ if $s_j = 0$ and on $(-2, 2)$ otherwise. Hence, the parameter $s_W$ controls the level of the layer sparsity: the smaller $s_W$, the higher the network's layer sparsity.

In concrete numbers, the input dimension is $d = 2$, the network widths are $p_2 = \cdots = p_l = 5$, the number of hidden layers is $l - 1 \in \{10, 25\}$, and the sparsity level is $s \in \{0.1, 0.3, 0.9\}$. Our settings and values represent, of course, only a very small part of possible networks in practice, but given the generality of our concepts, any attempt of an exhaustive simulation study must fail, and the simulations at least allow us (i) to corroborate our theoretical insights and (ii) to indicate that our concepts can be very useful in practice.

Datasets of 150 samples are generated; $n = 100$ of the samples are assigned to training and the rest to testing. The relevant measures for an estimate $\widehat{\boldsymbol{W}}$ of the network's parameters are the empirical mean squared error

$$\widehat{\text{mse}} \equiv \widehat{\text{mse}}[\widehat{\boldsymbol{W}}] := \frac{1}{|\mathcal{T}|} \sum_{(y, \boldsymbol{x}) \in \mathcal{T}} \left( y - f_{\widehat{\boldsymbol{W}}}[\boldsymbol{x}] \right)^2 ,$$

over the test set $\mathcal{T}$ with cardinality $|\mathcal{T}| = 50$ and the level of sparsity among the hidden layers

$$\hat{s} \equiv \hat{s}[\widehat{\boldsymbol{W}}] := \left| \left\{ j \in \{1, \ldots, l-1\} : h^{\text{L}, j}[\widehat{W^j}] \neq 0 \right\} \right| .$$

Reported are the medians (and third quantiles in paranthesis) over 30 simulation runs for each setting.

### 3.1.2 METHODS

Our first method (SLS) is a standard least-squares complemented with the layer regularizer (5) in Lagrange form $\widehat{\boldsymbol{W}}$. The baseline for this estimator is vanilla least-squares (LS). Since our estimator—in contrast to least-squares—allows for merging layers, we can also complement it with our refitting scheme of Section 2.4 (FLS). The baseline for our refitted estimator is the least-squares estimator that "knows" the relevant layers beforehand (ILS), that is, a least-squares on the relevant layers $\mathcal{V}_\mathcal{S}[\boldsymbol{W}]$ with $\boldsymbol{W}$ the true parameter—see Example 2. The latter estimator cannot be used in practice, but it can serve as a benchmark here in the simulations.

The objective functions are optimized by using mini-batch gradient descent with batch size 10, learning rate $10^{-2}$, and number of epochs 200 (for $l = 10$, $s_W = 0.1, 0.3$), 300 (for $l = 10$, $s_W = 0.9$), 400 (for $l = 25$, $s_W = 0.1$), and 500 (otherwise). The tuning parameters $(r^{\text{L}})_j$ are 0.2 (for $l = 10$, $s_W = 0.1$), 0.12 (for $l = 10$, $s_W = 0.3$), 0.07 (for $l = 10$, $s_W = 0.9$), and 0.05 (for $l = 25$).

### 3.1.3 RESULTS

The numerical results show that our layer-regularized version SLS can improve on the prediction accuracy of the standard least-squares LS considerably ($\widehat{\text{mse}}$-columns of the first and second rows in Table 1). The results also show that the refitting in FLS can improve the prediction accuracy further, and that refitted estimator can rival the infeasible ILS in terms of prediction ($\widehat{\text{mse}}$-columns of the third and fourth rows). The results finally show that the layer regularization can detect the correct number of layers ($\hat{\text{s}}$-columns of the third and fourth rows). In summary, our layer-regularized estimator outmatches the standard least-squares, and the refitted version of our estimator rivals the infeasible least-squares that knows which are the relevant layers beforehand—both in terms of prediction accuracy and sparsity. Hence, layer regularization can condense networks effectively.

The results also reveal that the prediction accuracies of our estimators increase with $\text{s}_\text{W}$ decreasing ($\widehat{\text{mse}}$-columns across different $\text{s}_\text{W}$). This trend is expected: the higher the layer sparsity, the more layer-regularization can condense the networks. This behavior is confirmed in the sparsities ($\hat{\text{s}}$-columns across different $\text{s}_\text{W}$). In other words, the layer regularization is adaptive to the true layer sparsity.

The tuning parameters $(r^\text{L})_j$ have been calibrated very roughly by hand. We expect that a more careful calibration of $\boldsymbol{r}^\text{L}$ based on cross-validation, for example, accentuates the positive effects of layer sparsity even further. But since our goal in this section is a general proof of concept for layer sparsity rather than the optimization of a specific deep learning pipeline, we do not pursue this further here.

The tuning parameters of the descent algorithm, such as the batch size, number of epochs, learning rate, and so forth, have also been calibrated very roughly by hand. An observation is the fact that all methods can sometimes provide accurate prediction if the number of epochs is extremely large, but our layer-regularized methods SLS and FLS generally lead to accurate prediction after much less epochs than their unregularized counterpart LS. This observation indicates that layer regularization also impacts the algorithmic aspects of deep learning beneficially.

## 3.2 REAL DATA

We now turn to real data. Specifically, we consider subsamples of `MNIST` (LeCun et al., 1998), `FashionMNIST` (Xiao et al., 2017), and `KMNIST` (Clanuwat et al., 2018).

### 3.2.1 SETUP

For each of the three data examples, the training data consists of $n = 10\,000$ images sampled uniformly at random; the test data also consists of $10\,000$ images. The network consists of $l = 10$ fully-connected layers; the hidden layers have width $p_2 = \cdots = p_l = 50$. While fully-connected networks have been outperformed image classification by other, more intricate pipelines, they still provide decent results (even on the subsetted data) and are perfectly suited for illustrating our ideas.

The baseline is cross-entropy (CE), which is LS but with the least-squares loss replaced by the cross-entropy loss. Our layer-sparse method is refitted cross-entropy (FCE), which is the pendant of FLS, that is, cross-entropy with additional layer-sparse regularization and refitting.

The objective functions are optimized by using mini-batch gradient descent with batch size 100, learning rate $10^{-3}$, and number of epochs 100. In line with theoretical considerations (Lederer & Vogt, 2020), the tuning parameters are set to $(r^\text{L})_j = \ln[\overline{p}]/\sqrt{n}$ with $\overline{p}$ the total number of network parameters. The performances are measured in terms of average classification accuracies (denoted those by $\widehat{AC}$) and level of sparsity among the hidden layers $\hat{\text{s}}$.

### 3.2.2 RESULTS

The results are summarized in Table 2. Similarly as before, we find that layer-sparse regularization can reduce the number of layers while retaining the classification accuracy or even improving it.

To highlight the features of layer sparsity more, we also look at the training losses and testing accuracies over the course of the optimization; Figure 2 contains these data averaged over 20 draws of the `MNIST` data. We find that both the initial estimator as well as the refitted version eventually

|        | MNIST |  | FashionMNIST |  | KMNIST |  |
|--------|-------|-----|--------------|-----|--------|-----|
| Method | $\widehat{AC}$ | ŝ | $\widehat{AC}$ | ŝ | $\widehat{AC}$ | ŝ |
| CE     | 0.92  | 10  | 0.84         | 10  | 0.76   | 10  |
| FCE    | 0.92  | 3   | 0.84         | 3   | 0.78   | 3   |

Table 2: layer sparsity can reduce model complexity without sacrificing classification accuracy

achieve the same accuracies, but the refitted version can be trained much faster. These observations are commensurate with the observations in (Frankle & Carbin, 2019), who study refitting of connection-sparse networks.

In contrast to (Frankle & Carbin, 2019), however, we could not find any benefits of keeping the initializations of the original network: both the training and the accuracy curves remain virtually the same. This observation might indicate that layer sparsity might be particularly robust under refitting.

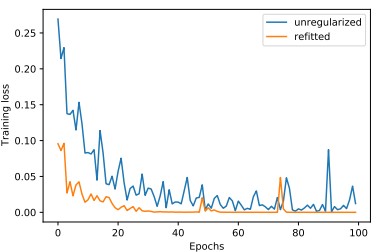 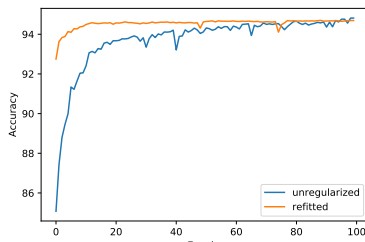

Figure 2: compressed layer-sparse networks can be trained faster than the original networks

## 4 DISCUSSION

We have shown that layer sparsity can compress layered networks effectively both in theory (Section 2) and practice (Section 3).

We have also shown that refitting with respect to layer sparsity can facilitate the optimization of the network. Refitting has a long-standing tradition in high-dimension statistics—see Lederer (2013), for example—but has been applied in deep learning only recently (Frankle & Carbin, 2019). Our research supports the usefulness of refitting in general, and it demonstrates that layer sparsity is particularly suited for refitting.

Related concepts such as ResNets add complexity to the network descriptions. This makes these networks not only unfit for refitting but also very hard to analyze statistically. Layer sparsity, in contrast, simplify network architectures and seem amenable to statistical analyses via recent techniques for regularized deep learning (Taheri et al., 2020). Statistical theory for layer sparsity, therefore, seems a feasible goal for further research.

In summary, layer sparsity complements other notions of sparsity that concern individual connections or nodes. All of these concepts can help to fit networks that are efficient in terms of memory and computations and easy to interpret.

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

# A  TUNING-PARAMETER CALIBRATION

Regularizing layer sparsity involves the tuning parameters $(r^{\mathrm{L}})_j$. Such tuning parameters are integral to regularization in deep learning and in machine learning more generally. In sparse linear regression, there has been some progress for developing theories for calibrating these parameters (Bien et al., 2019; Chichignoud et al., 2016; Taheri et al., 2016). In sparse deep learning, however, theories for tuning-parameter calibration are missing completely.

In the real-data analysis of Section 3.2, we have used a theory-inspired tuning parameter. The goal of this section here is to give more insights into the calibration of the tuning parameter. Figure 3 shows the accuracies of refitting with different number of hidden layers and locates the tuning parameters selected by our approach and by cross-validation, that is, training/validation based on 10 000 training and validation samples. The results are averaged over 20 draws of `MNIST` data as described earlier but only over 5 epochs each for illustration. The plot shows the expected upside-down-U-shape of the accuracies, which reflects the trade-off between variance (many layers/small tuning parameters) and bias (few layers/large tuning parameters). The plot shows that cross-validation can even improve the impact of layer sparsity further for a small number of epochs. (As illustrated in the right panel of Figure 2, tuning parameters become—as far as accuracy is concerned—less important for large number of epochs.)

Tuning-parameter calibration remains a challenge not only here but in deep learning much more generally. But our observations in this section and the main body of the paper demonstrate that layer-sparse regularization can improve deep-learning pipelines substantially with data-adaptive schemes such as cross-validation as well as with a simple, theory-inspired choice of the tuning parameter.

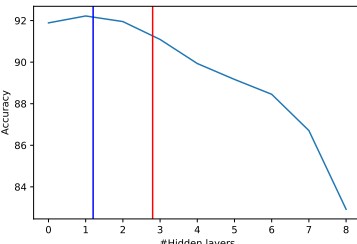

Figure 3: the accuracy as a function of the number of hidden layers has a typical upside-down-U-shape; the cross-validated tuning parameters (blue) improve the theory-inspired tuning parameters (red) further

