# OpenReview forum: "LAYER SPARSITY IN NEURAL NETWORKS"
_ICLR.cc/2021/Conference — Reject_

### Official Review · AnonReviewer1 · 2020-10-27
**Good ideas with poor experimental evaluation**

**Rating:** 4
**Confidence:** 5

**Review:**

The paper introduces a new, interesting definition of "sparsity" in a deep network. It penalizes as a single group the positive values in a layer. When combined with an activation function such as ReLU, they show that this allows to remove entire layers by merging two adjacent weight matrices.

The paper is well written and relatively easy to follow. The mathematical notation could be simplified (especially the difference between W and V).

However, the experimental results are definitely below what one expects in a deep learning paper today. The paper only considers a very simple, artificial setup with "Datasets of 150 samples" and tuning parameters "calibrated very roughly by hand". In my opinion this is not sufficient to show that a method is useful in a realistic context.

The paper should provide some discussion of the relation with recent literature on the lottery ticket hypothesis, especially when introducing its retraining procedure.

---

> ### Author Response · Authors · 2020-11-21
> **Thank you for your input**
>
> Dear Reviewer,
>
> Thank you for the positive feedback but also for the valuable suggestions for improvement.
>
> 1. Experimental results: Despite this not being an applied paper, we definitely agree that further numerical evaluations are of interest. Motivated by the reviews, we have now added a simple real-data analysis. We think that this analysis illustrates the features of layer sparsity nicely and sets the course for further numerical investigation.
>
> 2. Lottery tickets: Retraining predates lottery tickets considerably, but we absolutely agree that a discussion of this line of literature fits the paper extremely well. We have now added writing to Sections 2.4 (refitting), 3.1.1 (real-data analysis), and 4 (discussion).
>
> Again, thank you for the insightful input.

---

### Official Review · AnonReviewer2 · 2020-10-28
**Interesting idea, but maybe that's not enough?**

**Rating:** 6
**Confidence:** 3

**Review:**

__how I would summarize the paper.__
The paper gives an interesting new paradigm of the neural network compression called the *layer sparsity*. The paper builds on the (somewhat underappreciated) observation of Barron&Klusowski that positive weight parameters of multiple layers can be aggregated and reparametrized through the positively homogeneous activation functions. Based on the observation, the paper designs a regularizer that enhances the possibility of such aggregation by regularizing the negative parts only. Whenever successfully regularized, the number of layers in the model can be reduced without sacrificing too much performance, as empirically verified; actually, using the regularizer per se helps reducing the test mean-squared error.

__review tl;dr.__
While the idea itself is quite interesting, I believe that there should be more practical evidence and algorithmic extension of the framework, for the idea to fully bloom.

__what I like.__
The underlying idea itself is quite interesting (somewhat reminiscent of the [lookahead pruning](https://openreview.net/forum?id=ryl3ygHYDB)), and the design of the regularizer makes a perfect sense. Also, potential practical impact of the notion of layer sparsity seems to be big, as reducing the number of layers has a more straightforward practical benefit, in terms of reducing the inference flops/time.

__what I think is missing.__
Model compression became an important research area after deep, convolutional networks gained popularity. Perhaps this is why most of the works on network pruning focus on evaluating their algorithms on compressing deep convolutional networks trained on real (as opposed to synthetic) datasets. To fully persuade the benefit of the proposed algorithm, I think there should be (1) an extension of the algorithm to the aggregation of convolutional layers, which may lead to a larger kernel widths, which is okay, (2) an empirical validation on larger models trained on non-synthetic datasets.

A minor concern is that the refitting step does not seem particularly novel, as the retraining step is already typical in model compression literature.

__a question.__
Would it be possible to compress the layer which does not perfectly satisfy the criterion (with a slight abuse of terminology) at "page 6, penultimate line," but only approximately 0?

---

> ### Author Response · Authors · 2020-11-22
> **Valuable Input**
>
> Dear Reviewer,
>
> Thank you for pointing out the large potential of our concepts and for the additional input. We discuss this input in the following.
>
> 1. CNNs: Indeed, layer sparsity applies far beyond fully-connected layers; in particular, layer sparsity can be applied to convolutional layers as well. We have now written this much more clearly in the new Section 2.5---thank you for raising this question!
>
> 2. Real Data: We have now added a real-data analysis in Section 3.2, which---as we think---supports our claims very well. (An extension to convolutional layers is underway; the preliminary results align exactly with the ones in the fully-connected case.)
>
> 3. Refitting: We have now detailed the writing about refitting in Section 2.4.
>
> 4. Approximate layer sparsity: Indeed, we could think about compressing networks when layer sparsity holds only approximately. However, at this point, we think that a better strategy for more aggressive compression is to choose larger tuning parameters. The advantage is the fact that Theorem 1 then holds exactly. Still, some relaxation can avoid numerical issues, and approximate layer sparsity might be an interesting topic to look at more generally.
>
> Thank you again for your valuable and insightful comments.

---

### Official Review · AnonReviewer3 · 2020-10-28
**Interesting method, but insufficient empirical analysis**

**Rating:** 5
**Confidence:** 3

**Review:**

##########################################################################

Summary:

The paper proposes a regularizer enforcing a novel form of sparsity that authors call "layer sparsity". Under certain conditions on layer weights, two consecutive layers in a deep neural network (with certain nonlinear activation functions) can be represented exactly as a single layer. The authors proposed a regularizer that can lead to such layer collapse thus resulting in shallower and more compact models.


##########################################################################

Reasons for score:

Overall, I vote for rejecting this paper. In my opinion, the proposed regularizer is quite simple and intuitive, but is at the same time novel and could potentially be useful in practice. However, the proposed empirical studies shed very little light on how effective this method can be in practical and even remotely realistic circumstances. I am afraid that the current empirical evaluation is not sufficient.


##########################################################################

Pros:

1. The paper is clearly written. The core idea of the layer sparsity regularizer is introduced and explained with a sufficient level of mathematical rigor.

2. The proposed technique could become a useful addition to the deep learning practitioner's toolbox.


##########################################################################

Cons:

1. While the experiments described in the paper are a reasonable first step in exploring the effect of adding the proposed regularizer, more empirical exploration (particularly with realistic and practical models) might be necessary for a sufficiently thorough analysis. Currently there is just not enough information to judge the effectiveness of this method in any practical circumstances.

##########################################################################

Questions during rebuttal period:

Please address and clarify the cons above. (I will update the score depending on the authors reply.)

I also have a question (and doubts) about Theorem 1 and its proof in Appendix A.

To illustrate my confusion consider a simple model with ReLU activations $f(x)$.
Then $o_i = f\left(\sum_j w_{ij} f\left(\sum_k w_{jk} x_k\right)\right)$ can be rewritten as $o_i = \sum_j f\left(\sum_k w_{ij} w_{jk} x_k\right)$ assuming that all $w_{ij}\ge 0$.
Even after reading Appendix A, I do not understand how dependence on index $j$ inside $f(\cdot)$ can be factored out.
In other words, I do not understand how this expression can be written as $f^{j,j+1}(V^{j,j+1}z)$ with $V^{j,j+1}\in \mathbb{R}^{p_j \times p_{j+2}}$ and $f^{j,j+1}:\mathbb{R}^{p_j}\to \mathbb{R}^{p_j}$.
I think I can even come up with simple numerical examples illustrating this point (all we need to do is make sure that the argument of $f$ is negative for one particular $j$ thus making this term vanish completely in the original expression, but still be present in the result of Theorem 1).

In a long derivation in Appendix A, in a transition from line 2 to line 3, the expression with an index $m$ fixed externally is seemingly replaced with a scalar product involving a summation over this index (at which point it disappears altogether from inside the activation function). It is entirely possible that I do not understand something trivial, but I would ask the authors to explain this transition.

#########################################################################

Other minor typos:

1. A. Barron and J. Klusowski 2018 reference is repeated twice (there must be two records for it in the bibliography file).

#########################################################################

Post-rebuttal.

I would like to thank the authors for their reply, which addressed some of my questions. While I now agree with the main theoretical result when applied to a ReLU nonlinearity, this of course also reduces the area of applicability of the proposed technique. I am also happy to see additional empirical results, which I believe will be the key to making this paper much stronger (since the current theoretical result is actually quite straightforward when applied to the ReLU nonlinearity). But I think the results are still a bit insufficient to make this submission sufficiently strong. One of my concerns is the final accuracy in Figure 2. We can see that the unregularized model surpasses the accuracy of the refitted model and could potentially get higher (or even much higher) should it not have been cut at <100 epochs. I will be excited to see an updated and improved version of this paper in the future, but in my opinion, the current version still needs a bit of work and is not entirely convincing.

---

> ### Author Response · Authors · 2020-11-19
> **Thank you for the suggestions**
>
> Dear Reviewer,
>
> Thank you for the careful reading of our paper and for the valuable input. In short, we agree with all of your comments, and we see much potential for this contribution.
>
> In view of the current trend toward deeper and deeper networks, we expect depth regularization to become increasingly important. We thus agree that layer sparsity can become a widely-used concept in practice.
>
> On the other hand, we agree with your input about the extent of the empirical study. But we also think that this issue can be remedied in rather straightforward ways. As a start, we have added a standard MNIST-type analysis in Section 3.2 of the new version. The purpose of this section is to extend our empirical results to real data and to classification. We again find very encouraging performances of layer sparsity.
>
> Thank you for reading our theory in detail. In brief: We now think it is best to focus on a simple ReLU setting to fix ideas, because this avoids digression and emphasizes the simplicity of the idea further. (This should also remove your concerns about the proof.)
>
> We will keep working on the paper, especially on the real-data application, but we hope that the new version alleviates some of your concerns already.

---

### Official Review · AnonReviewer4 · 2020-10-29
**Major Revision Needed**

**Rating:** 5
**Confidence:** 3

**Review:**

This paper proposes a new notion of layer sparsity for neural networks that aims at simplifying network architectures and reducing the number of parameters. A type of regularizers has been introduced to encourage this specific structure.

Overall the paper is well written and the idea is interesting. It pushes the network towards less nonlinear layers if not needed. Does the method apply to scenarios where there are bias terms between the layers?

The numerical results are not sound. Using a fixed number of epochs in each setting, we have no idea of the convergence status for different methods. The authors claim in the end of the results section that all methods can sometimes provide accurate prediction if the number of epochs is extremely large. Does this happen to the numerical examples presented here? Plotting some training curves might help readers understand more of the behavior.

It is fairly standard practice to tune the regularization parameters based on cross-validation or an independent validation set. The authors should include this for the completion of the manuscript, even if the numerical experiment might serve as a proof of concept.

It would be better to include comparison with other sparsity-inducing methods such as connection sparsity and node sparsity. Also, it would be interesting to see the performance in other settings than the most favorable one, for example on some real data sets.

---

> ### Author Response · Authors · 2020-11-23
> **Thank you for the valuable input**
>
> Dear Reviewer,
>
> Thank you for your insightful comments. We are glad you like the idea, and we also appreciate your further suggestions. We have included all of these suggestions in the new version of the paper.
>
> 1. Bias: Indeed, bias terms can be handled readily. We have now clarified this in Section 2.5 on Page 6.
>
> 2. Training curves: We have now included training curves in Figure 2 on Page 9.
>
> 3. Tuning parameters: We have now provided insights on the calibration of the tuning parameters in the Appendix on Page 12.
>
> 4. Real-data analysis: We have now included a real-data analysis on Pages 8–9. However, we do not think of layer sparsity as a competitor of connection or node sparsity: instead, these three notions nicely complement each other. We have now made that more clear at the end of Section 2.3 on Page 5. But since connection and node sparsity have been discussed previously, we focus sharply on layer sparsity in our numerical study.
>
> Thank you again for the valuable feedback.

---

### Author Response · Authors · 2020-11-23
**Summary**

We thank the reviewers for their valuable feedback. It seems that our idea of layer sparsity is unanimously appreciated and believed to have large potential. The only major limitation that seems to remain is the absence of large-scale validations. We agree that large-scale validations could increase the immediate impact of the paper, and that they are useful in deep learning in general. But we also believe that the field benefits from being open to novel ideas that are not directly accompanied by such massive computations.

Again, we appreciate the input and the interactions with the reviewers.

---

### Decision · Program_Chairs · 2021-01-07
**Final Decision**

**Decision:**

Reject

**Comment:**

Reviewers agree that the idea of layer wise regularization is interesting and is in line with many efforts in the optimization realm to specialize in the training procedure and the learning rate to each layer.  Given the depth of some state of the art neural networks, efficiency is at stake and the idea brought up in this paper naturally falls into that.  While the theoretical result in Theorem1 is sound and clear, an extended result on the impact of such « merge » and « layer skipping » on the overall predictions of the algorithm can be well appreciated. The overall goal of network compression should remain to reduce drastically the network size, and thus the training time (energy consumption etc...), while keeping a relatively good prediction accuracy (at least of the same order). Being able to back this with theory (and of course experiments) is crucial.   Reviewers also pointed out that the empirical evaluations were not sufficient for ICLR. For example, there are no enough comparisons with existing algorithms and there should be more experimental results based on real datasets. Although the rebuttals did help clarify some of the issues raised by the reviewers, overall this paper does not seem to meet the bar to be accepted.